# Astaxanthin Ameliorates Ischemic-Hypoxic-Induced Neurotrophin Receptor p75 Upregulation in the Endothelial Cells of Neonatal Mouse Brains

**DOI:** 10.3390/ijms20246168

**Published:** 2019-12-06

**Authors:** Min-Hsun Kuo, Hung-Fu Lee, Yi-Fang Tu, Li-Hsuan Lin, Ya-Yun Cheng, Hsueh-Te Lee

**Affiliations:** 1Taiwan International Graduate Program in Molecular Medicine, National Yang-Ming University and Academia Sinica, Taipei 11529, Taiwan; rickuo626@gmail.com; 2Institute of Anatomy and Cell Biology, National Yang-Ming University, Taipei 11202, Taiwan; eternalisr@hotmail.com (L.-H.L.); lab.j512.1@gmail.com (Y.-Y.C.); 3Department of Neurosurgery, Cheng Hsin General Hospital, Taipei 11202, Taiwan; ufae0073@ms7.hinet.net; 4Department of Pediatrics, National Cheng Kung University Hospital, Colleague of Medicine, National Cheng Kung University, Tainan 70403, Taiwan; nckutu@gmail.com; 5Taiwan International Graduate Program in Interdisciplinary Neuroscience, National Yang-Ming University and Academia Sinica, Taipei 11529, Taiwan

**Keywords:** hypoxic-ischemia-reperfusion, brain, endothelial cells, p75NTR, AXT

## Abstract

Ischemic stroke is a leading cause of human death in present times. Two phases of pathological impact occur during an ischemic stroke, namely, ischemia and reperfusion. Both periods include individual characteristic effects on cell injury and apoptosis. Moreover, these conditions can cause severe cell defects and harm the blood-brain barrier (BBB). Also, the BBB components are the major targets in ischemia-reperfusion injury. The BBB owes its enhanced protective roles to capillary endothelial cells, which maintain BBB permeability. One of the nerve growth factor (NGF) receptors initiating cell signaling, once activated, is the p75 neurotrophin receptor (p75NTR). This receptor is involved in both the survival and apoptosis of neurons. Although many studies have attempted to explain the role of p75NTR in neurons, the mechanisms in endothelial cells remain unclear. Endothelial cells are the first cells to encounter p75NTR stimuli. In this study, we found the upregulated p75NTR expression and reductive expression of tight junction proteins after in vivo and in vitro ischemia-reperfusion injury. Moreover, astaxanthin (AXT), an antioxidant drug, was utilized and was found to reduce p75NTR expression and the number of apoptotic cells. This study verified that p75NTR plays a prominent role in endothelial cell death and provides a novel downstream target for AXT.

## 1. Introduction

About 795,000 people suffer from strokes annually in the United States, making it the leading cause of disability [1]. The majority of strokes are ischemic strokes, which consist of two distinct phases of pathological impact, namely, ischemia and reperfusion [2]. The ischemic phase of a stroke results from the loss of cerebral blood flow and nutrition due to mechanical blockage of a blood vessel via a thrombus or an embolus [1]. Though blood reperfusion is necessary for tissue survival, it also contributes to additional tissue damage [3,4,5,6,7]. Injuries of reperfused tissue have also been related to the excessive production of reactive oxygen species in said tissue [8,9,10]. Oxidative stress and neuroinflammation are significant contributors to blood-brain barrier (BBB) damage because they can disrupt endothelial tight junction proteins, resulting in increased paracellular permeability [11,12]. Therefore, the BBB is considered the most critical target of ischemia-reperfusion attack [13,14]. Neurovascular unit dysfunction can result from BBB hyperpermeability, in association with oxidative stress and neuroinflammation in selected neurological disorders, such as strokes, epilepsy, Alzheimer’s disease, traumatic brain injury, and multiple sclerosis [15]. Tight junction proteins are the most crucial factors for modulating barrier integrity, including the claudin family, occludin, and zonula occludens-1 (ZO-1), which are present in cerebral vascular endothelial cells [16]. ZO-1 and claudin-5 are the most important components for cell barrier integrity and stability [17]. It has been suggested that neurovascular unit dysfunction after an ischemia-reperfusion injury is regarded as the dysregulation of tight junction proteins [16,17,18].

In inflammatory diseases, the nerve growth factor (NGF), in the neurotrophin family, has been conclusively shown to be associated with inflammatory state [19]. The nerve growth factor regulates the differentiation, growth, and survival of neurons in the central nervous system (CNS), and the p75 neurotrophin receptor (p75NTR) is one of kind of NGF receptor [20]. This receptor shows low binding affinity with NGF, and previous studies shown that uncleaved pro-NGF can bind to p75NTR with high affinity, and thus causes cell death [21]. However, there are very few reports regarding the expression and the functional role of p75NTR in brain endothelial cells. Previous research has focused on the damage of neurons under stress induced by ischemia-reperfusion injury, but has not taken endothelial damage into account.

Astaxanthin (3,3′-dihydroxy-β, β′-carotene-4,4′-dione, herein AXT) is a well-known xanthophyll carotenoid of predominantly marine origin [22,23,24]. It has been shown to have up to 10 times higher antioxidant activity than other carotenoids [25]. AXT’s many biological functions have also been reported, including anti-inflammatory, anticarcinogenic activity, and cardioprotective effects [26,27,28]. Although AXT’s protective role in the cerebral microvascular endothelial cell against oxidative damage has been established, the mechanisms underlying this protection have not been investigated.

In this study, we investigated the role of p75NTR in cerebral microvascular endothelial cells under stress induced by ischemia-reperfusion injury and focused on the BBB hyperpermeability resulting from the dysregulation of tight junction proteins. We found that p75NTR protein expression was upregulated by ischemia-reperfusion injury in endothelial cells in both in vivo and in vitro models. Also, tight junction protein expressions were decreased, resulting in increased dextran permeability. Next, we investigated whether p75NTR is involved in the cytoprotective effect of AXT in terms of oxidative stress induced by ischemia-reperfusion injury in endothelial cells. Our results showed that p75NTR protein expression was downregulated and endothelial cell survival was increased with AXT treatment after ischemia-reperfusion injury. Together, we have demonstrated that p75NTR in endothelial cells under ischemia-reperfusion injury lead to apoptosis, and the protective effect of AXT is accomplished via the downregulation of p75NTR.

## 2. Results

### 2.1. Neonatal Mice Brains Were Severely Damaged after Ischemic-Hypoxic Reperfusion

To study the ischemia-reperfusion effects on mice, we used the experimental paradigm adapted from the Rice–Vannucci hypoxic-ischemic (HI) model [29]. The experimental design and time scheme used is illustrated in Figure 1A (see Section 4.2 for a full description of the procedure). The left common carotid arteries of the mice pups were ligated. After the surgery, the mice brains were observed by Nissl staining, and the precise extent of the injury was assigned a brain injury score, and the data of the intervention pups were compared with the sham group (Figure 1B). Interestingly, 3 h after HI-reperfusion, the p75NTR-positive cells were markedly present in the vessels of the ipsilateral cortex, which was determined via immunofluorescence analysis (Figure 1C). Altogether, these findings suggest that p75NTR in the blood vessels instead of the neuronal cells may be involved in the initial stages of HI-reperfusion stress.

### 2.2. AXT Treatment Effectively Decreased HI-Induced Brain Injury in Neonatal Mice

Next, we investigated the impact of AXT in HI-induced brain injury in mice. At P7, 30 min before ligation surgery, we pretreated the mice with the vehicle and AXT (40 mg/Kg and 80 mg/Kg, respectively, Figure 2A). Our data indicate that the brain injury region in mice pretreated with AXT (80 mg/Kg) was significantly rescued compared with the vehicle pretreatment group (Figure 2B). Furthermore, immunohistological analysis confirmed that AXT (80 mg/Kg) reduced p75NTR expression in the endothelial cells, which had fewer lesions (Figure 2C). These findings suggest that a single dose of AXT might potentially be a treatment for HI-induced brain injury via p75NTR expression reduction in endothelial cells.

### 2.3. Oxygen-Glucose Deprivation/Reperfusion Treatment Decreased the Cell Viability and Tight Junction Stability of bEnd.3 Cells

Next, we attempted to explain the neuroprotective effect of AXT on the BBB. It is known that endothelial cells play a part in the formation of the BBB and have a potent role in monitoring circulation. We created an in vitro model to verify our hypothesis. To mimic the BBB under conditions of injury resulting from mild ischemia-reperfusion, we established an appropriate model by utilizing the mouse brain microvascular endothelial cell line bEnd.3. The bEnd.3 cells were exposed to oxygen-glucose deprivation/reperfusion (OGDre) conditions for 12 h and reperfusion for 12 h (Figure 3A). Significant morphological alterations in the OGDre12/12 group were observed compared to the control group (Figure 3B). The cells viability and monolayer formation ability were reduced after OGDre (Figure 3B). The cell viability of the OGDre12/12 group was only about 63%, indicating severe cell death (Figure 3C). Moreover, the permeability of the monolayer endothelial cells increased dramatically after OGDre, as detected using FITC-dextran (Figure 3D). We also detected the expression of HIF1-α, a hypoxia-induced transcription factor, which was used to evaluate the hypoxic stress. Our results showed that HIF-1α expression level increased under OGDre compared to the control (Figure 3E). Next, the tight junction related proteins claudin-5 and ZO-1 were also enrolled to evaluate the tight junction of bEnd.3. OGDre induced a decrease of the protein level expressions in both ZO-1 and claudin-5 in bEnd.3 (Figure 3E). This evidence shows that, in the OGDre12/12 group, both hyperpermeability and the expression of tight junction proteins in bEnd.3 cells were decreased.

### 2.4. Apoptosis Was Induced by Oxygen-Glucose Deprivation/Reperfusion Injury

We used a TUNEL assay to confirm if cells were dead or not. There were no TUNEL positive cells in the control group, but the TUNEL positive cell number increased in the OGDre group (Figure 4A). The statistical analysis showed that OGDre induced a 31-fold increase in TUNEL positive cells in the bEnd.3 cells (Figure 4B). Moreover, oxidative stress was detected. As excessive production of reactive oxygen species (ROS) is known to be a significant contributor to BBB damage. Next, we investigated the ROS level with a DCFDA cellular ROS detection assay kit in our OGDre model. We found the ROS level had about a 60-fold increase following the stress induced by OGDre (Figure 4C). Together, these results suggest that OGDre treatment could trigger cell apoptosis potentially induced by oxidative stress.

### 2.5. AXT Attenuates Oxygen-Glucose Deprivation/Reperfusion-Induced Cell Survival and Apoptosis

Due to the role in reducing oxidative stress, we considered AXT as a potential treatment for OGDre-induced injury. The bEnd.3 cells were treated with two AXT dosages (10 μM and 20 μM) that conferred a dose-dependent resistance to the stress caused by OGD-reperfusion (Figure 5A,B). Additionally, morphological restorations were elevated in the group treated with 20 μM of AXT (Figure 5C). Also, after treatment with 20 μM of AXT during the reperfusion, TUNEL-positive cells decreased compared to the vehicle group under OGD reperfusion (Figure 5D). The statistical analysis showed that AXT induced about a 6.6-fold decrease in TUNEL positive cells in bEnd.3 cells (Figure 5E). Moreover, our data indicate that the treatment with AXT resulted in an inhibition of intracellular production of ROS. The 20 μM AXT pretreatment group revealed a 2-fold decrease compared to the vehicle group under OGD reperfusion (Figure 5F). Therefore, these findings suggest that AXT could restore the survival rate and increase oxidative stress resistance of bEnd.3 cells.

### 2.6. AXT Maintains the Tight Junction Stability of bEnd.3 Cells

After observing restorations of cell survival, we also treated bEnd.3 cells with 20 μM AXT during the reperfusion to evaluate the permeability and tight junction proteins. We found that the permeability decreased by about 1.8-fold by detecting FITC-dextran (Figure 6A). Western blotting data indicated that the tight junction related protein ZO-1 expression increased after the AXT treatment compared to the vehicle group (Figure 6B,C). A comparable increase was seen in claudin-5 expression (Figure 6B,D). Moreover, the expressions of ZO-1 also detected higher expression in the mice receiving AXT treatment (Figure 6E). Collectively, AXT’s effect on ODGre helps not only the cell to survive but also maintains the permeability resistance ability.

### 2.7. p75NTR is Expressed in Cerebromicrovascular Endothelial Cells with Enhanced Expression after OGD Reperfusion

p75NTR is a neurotrophic receptor that can regulate neuronal cell death. However, previously in this study (Figure 1C), it was determined that p75NTR is firstly highly expressed in endothelial cells under early-stage ischemia-reperfusion rather than only in neuronal cells. Therefore, we investigated the expression of p75NTR in bEnd.3s after OGDre. The p75NTR expression significantly increased under OGDre treatment compared to the control (Figure 7A,B). Subsequently, the AXT treatment in the reperfusion stage pressed the p75NTR expression (Figure 7C,D). These results suggest that p75NTR may play a critical role in injury induced by OGDre and could be pressed after AXT treatment.

### 2.8. p75NTR Is Required for Unstable Tight Junction and Survival in OGDre-Induced Injury

We wanted to determine whether p75NTR is required to make tight junctions stable under OGDre-induced injury. The p75NTR knockdown-stable bEnd.3 cells were established to test the possibility of this. The p75NTR knockdown cells increased endogenous ZO-1 expression (Figure 8A,B). This suggests that p75NTR may play a direct role for tight junction protein in bEnd.3 cells. Additionally, we found the p75NTR knockdown-stable bEnd.3 cells could survive better in OGDre-induced cell death (Figure 8C). Furthermore, we analyzed the related tight junction proteins in the p75NTR knockdown cells under OGDre stress, finding that p75NTR knockdown could increase both ZO-1 and the claudin-5 expression levels (Figure 8D) under OGDre-induced stress. Similarly, we found that TUNEL-positive p75NTR knockdown-stable cells notably decreased compared to the mock group under the OGD reperfusion condition (Figure 9A,B). This evidence suggests that p75NTR is required for maintaining tight junction stability and endothelial cell survival under OGDre stress conditions.

## 3. Discussion

In this study, we demonstrated the prominent role of p75NTR in brain capillary endothelial cells under OGDre-induced injury. The bEnd.3 cells used in our investigation after the ischemia-reperfusion and the FITC-dextran permeability test demonstrated the feasibility of this study’s model. In addition to endothelial cells, the BBB is composed of pericytes, astrocytes, neurons, and the extracellular matrix (ECM), which have been collectively redefined as the neurovascular unit [14]. Previous studies have indicated that the BBB under ischemia-reperfusion injury would lose integrin adhesion molecule expression and induce tight junction dysregulation in the endothelium [30]. Our data indicate that p75NTR expression of endothelial cells increases after OGDre stress, decreasing tight junction protein expression.

The endothelium is the first line of defense between the circulatory system and the brain. The endothelial cells of the BBB are distinguished from peripheral endothelial cells by their lack of fenestrations, minimal pinocytotic activity, and the presence of tight junction proteins [31]. In our study, we not only applied the in vivo model but also established an ischemia-reperfusion in vitro model by exposing endothelial cells to oxygen-glucose deprivation. Moreover, the data showed that mild ischemia-reperfusion stress reduced endothelial cell viability and FITC-dextran permeability increased following OGD-re stimulation, and increased FITC-dextran permeability was associated with decreased tight junction protein expression. We next found that the expression of p75NTR (a member of the tumor necrosis factor receptor superfamily associated with neuronal apoptosis) was enhanced by ischemia-reperfusion stimulation in endothelial cells and positively correlated with ROS production and endothelial cell death. Collectively, our results, suggest a regulatory role of p75NTR in endothelial cell tight junctions, permeability, and apoptosis after mild ischemia-reperfusion stimulation, and that p75NTR expression is ROS-dependent. Besides, we revealed p75NTR as a novel downstream factor for AXT antioxidant and anti-apoptotic function. p75NTR can promote either survival or death in neuron cells and modulates neurite outgrowth, depending on the operative ligands and expression of downstream signaling elements [32]. Most of the studies on p75NTR have focused on neuronal cells rather than endothelial cells. Moreover, our results indicate that the protein expression of p75NTR was markedly elevated in brain endothelial cells preceding neurons after ischemia-reperfusion treatment. In the current study, p75NTR was also upregulated and accompanied with decreased viability in the bEnd.3 cell after OGD-re injury. Together, these data suggest that p75NTR may be involved in a decline of brain endothelial cell viability via cell death following OGD-re injury.

We determined that the cell death after OGD-re injury in our model was apoptotic using a TUNEL assay. Apoptosis can been induced by several types of cellular stress stimuli, including oxidative stress [33]. Moreover, the excessive production of ROS has been observed and verified in oxidative stress-induced apoptosis [8,9,10]. Furthermore, in anticancer drug research, p75NTR has been shown to regulate apoptosis via the p38 mitogen-activated protein kinase (p38-MAPK) pathway [34]. Therefore, the p38-MAPK pathway is an appealing downstream target for p75NTR in our model, which requires future research for validation.

As previously reported, the excessive production of ROS is one of the leading causes of oxidative damage, and increased levels of ROS after OGD-re injury were also detected in our model [8,9,10]. ROS are essential determinants in cell signaling pathway regulation and are involved in proliferation, apoptosis, and senescence [35]. However, Mi et al. demonstrated that p75NTR provided a protective role against oxidative stress-induced apoptosis for PC12 cells [36]. Contradicting this notion, in our model, p75NTR exhibited a positive correlation with ROS after OGD-re injury, and both p75NTR expression and ROS production were positively associated with increased numbers of apoptotic endothelial cells. This discrepancy between Mi et al. and our model may be due to the differences in cell types and the generation of ROS. However, we found that endothelial p75NTR expression preceded neuronal p75NTR expression, and with the complexity of cell-cell interactions in the BBB, p75NTR-induced endothelial changes may impact the function of neuronal p75NTR. Additionally, claudins, occludins, and zonula occludens-1 are the proteins responsible for the enhanced barrier function in the BBB when compared to junction complexes in other tissue barriers [14]. Claudin-5 is typically expressed in brain endothelial cells and is readily detected in bEnd.3 as well [37,38]. Claudin-5 regulates the paracellular permeability of the BBB [17].

Furthermore, the claudin-5 gene-deficient transgenic mice show a loss of BBB integrity [37]. It has also been suggested that occludins play a vital role in the barrier function of tight junction proteins [39]. We focused on the delayed stage of ischemic induced injury. Since occludin expression is associated with early and acute responses of the BBB, it is irrelevant to the current study. ZO-1 is involved in the BBB tight junction formation and regulation since it serves as a bridge between transmembrane proteins and skeleton proteins [31]. Currently, there is no direct evidence for how p75NTR interacts with tight junction proteins. However, previous studies have demonstrated that tight junction proteins distinctively undergo proteolytic cleavage and are correlated with a disruption of the tight junction after the induction of apoptosis [40]. Moreover, we found that the ZO-1 expressions restored and maintained cell viability after p75NTR knockdown in our results (Figure 8). Together, p75NTR could play a potent role in exacerbating cell death induced by ischemic-hypoxic reperfusion. Although detail of the mechanism involved in p75NTR and these junction proteins has not been clarified, we have provided a potential target to study the relationships among those integrin adhesion molecules and p75NTR in future.

AXT, a xanthophyll carotenoid antioxidant, was used in our model to reduce the production of ROS induced by OGD-re injury. AXT has been shown to display elevated levels of antioxidant activity [25] and the ability to scavenge superoxide anion radicals [41,42,43]. Furthermore, AXT can cross the blood-brain barrier, unlike other antioxidants, including β-carotene and lycopene [44]. AXT is also soluble in lipids, so it can be incorporated into cell membranes [45]. Therefore, it is an ideal treatment in our model. AXT has also been reported to exert anti-inflammatory and anti-apoptosis functions [26,27,28]. It can inhibit ultraviolet-induced apoptosis by decreasing the expression of pro-inflammatory mediators, including IL-1β and TNF-α [46], and also inhibit oxidative stress-induced apoptosis by decreasing the production of ROS [47], which is consistent with our results. Our results indicate that AXT-attenuated apoptosis is induced by OGD-re stimulation via inhibition of the expression of p75NTR. We suspect that p75NTR may be a novel downstream target of the anti-apoptotic function of AXT.

## 4. Materials and Methods

### 4.1. Reagents and Assay Kits

For this study, the crystal form of AXT (LemnaRed^®^ Crystal, provided by Lemnaceae Fermentation Inc., Tainan, Taiwan) was extracted and purified from engineered *E. coli* biomass, which was propagated under optimized aerobic conditions [48,49]. Assays used include the DCFDA Cellular Reactive Oxygen Species Detection Assay Kit (Abcam, Burlingame, CA, USA), and the HyECL Western Chemiluminescent Kit (HyECL Biotechnology Inc., Taipei, Taiwan). For detecting and quantifying apoptosis at the cellular level, the study used the In Situ Cell Death Detection Fluorescein Kit from Roche Diagnostics (Rotkreuz, Switzerland).

### 4.2. A Mice-Pup HI Model and AXT Treatment

This study was approved by and followed the guidelines of the Institute Animal Care and Use Committee (IACUC) of the National Yang-Ming University (project identification code: 1041109, approved on 5 November 2015). The experimental paradigm was adapted from the Rice–Vannucci HI model [29]. Neonatal male C57BL/6J mice pups (5–7 per dam) were housed with their dams in a temperature- and humidity-controlled colony room until they were sacrificed on the postpartum day (P) 14 to collect tissue for analysis. For the HI groups, P7 mice pups were anesthetized with 4% isoflurane (room air balance), and the left common carotid artery was surgically exposed and permanently ligated with 5–0 surgical silk. After surgery, the pups were returned to their dams to recover for 60 min before hypoxia. The pups were then placed in a transparent, airtight chamber (A-Chamber with a NexBiOxy Oxygen Controller, Taiwan) perfused with a humidified gas mixture containing 8% oxygen balanced with 90% nitrogen at 37 °C for 30 min. After hypoxia was completed, the mice pups were returned to their cage [50]. After the process was complete, the HI mice were sacrificed at the P14 for analysis. For the sham group, P7 mice were under the surgery without ligation of left common carotid and also sacrificed at the P14 to analyze. For the AXT treatment, the vehicle (olive oil; Olitalia S.r.l, Italy) or AXT 40 and 80 mg/kg groups were executed, as in the previous study [33], where AXT was administrated to the P7 mice pups with vehicle and AXT intraperitoneal injection 30 min before the HI treatment and also sacrificed at P14 for analysis.

### 4.3. Cell Culture and OGD Reperfusion Conditions

Mouse immortalized cerebral endothelial (bEnd.3) cells (American Type Culture Collection) were cultured as described previously [51]. In brief, the bEnd.3 cells were maintained at a density of 2 × 10^5^ cells per well in Dulbecco’s modified eagle medium (DMEM) (Coring Inc., Corning, NY, USA), supplemented with 10% fetal bovine serum (FBS), and 1% penicillin/streptomycin in a 37 °C incubator with a humidified atmosphere of 5% CO_2_ and 95% air. After reaching about 80–90% confluence, the cells were passaged using trypsin-EDTA. For the stably cloned p75NTR knockdown cells, we purchased the p75NTR shRNA (Cloned ID: NM_033217) from the RNAiCore (Academia Sinica, Taipei, Taiwan). After treatment with knockdown shRNA, the cells were selected by puromycin, 2 μg/mL, for 1 month, as well as the mock groups. To induce oxygen glucose deprivation (OGD), the bEnd.3 cells were washed twice with glucose-free DMEM (OGD medium), switched to OGD medium with 2% FBS, and exposed to a 1% oxygen environment for 12 h in a hypoxia chamber (NexBiOxy Inc., Taipei, Taiwan) inside a humidified modular incubator (Forma Scientific, Marietta, OH, USA). The bEnd.3 cells underwent OGD for 12 h OGD was terminated by returning the cell culture to a normoxic condition and replacing the medium with normal culture complete medium at 37 °C for 12 h.

### 4.4. AXT Administration in Cell Culture

The stock of AXT was prepared by dissolving 5% crystal-oil in dimethyl sulfoxide (DMSO, BioShop) at a 10 mM concentration. AXT was applied to the bEnd.3 cells at final concentrations of 10 and 20 μM during the reperfusion phase.

### 4.5. Cell Viability

Analysis of cell survival was performed by an MTT assay using 3-(4,5-dimethylthiazol-2-yl)-2,5-diphenyltetrazolium bromide (MTT), which was added to a final concentration of 50 μg/mL and incubated at 37 °C with a humidified atmosphere of 5% CO_2_ and 95% air for 4 h. After incubation, the medium was carefully aspirated to avoid formazan detachment. Here, 50 μL/mL of dimethyl sulfoxide (DMSO) was added to each well and mixed thoroughly by pipetting. After 10 min of solubilization in the culture incubator, the sample was mixed, and absorbance was read at 540 nm using a multimode microplate reader (Infinite 200, TECAN, Tecan Trading AG, Switzerland).

### 4.6. Western Blot Analysis

Samples of 40 μg were resolved using 10% sodium dodecyl sulfate-polyacrylamide gel electrophoresis and blotted electrophoretically to polyvinylidene fluoride membranes. In brief, the membranes were incubated with primary antibodies, and immunoreactivity was detected using a horseradish-conjugated secondary antibody and visualized using enhanced chemiluminescence. The following primary antibodies were used: Mouse anti-HIF-1α antibody (1:1000, abcam, Burlingame, CA, USA ), rabbit anti-p75NTR monoclonal antibody (1:1000, Cell Signaling and Proteintech), rabbit anti-claudin-5 polyclonal antibody (1:500, Santa Cruz Biotechnology, Inc., Santa Cruz, CA, USA), goat anti-ZO-1 polyclonal antibody (1:500, Santa Cruz Biotechnology, Inc., Santa Cruz, CA, USA ), and, rabbit anti-β actin monoclonal antibody (1:500, Sigma Aldrich, St. Louis, MO, USA). For densitometry, data were normalized for the actin expression.

### 4.7. Dextran for Endothelial Permeability Assay

The flux of FITC-conjugated dextran (FITC-dextran, 40 kDa, Sigma) across the bEnd.3 cell monolayer was used to measure the paracellular permeability. In brief, 24-mm diameter polyethylene terephthalate hanging cell culture inserts (0.4 μm pore size, Millipore, Burlington, MA, USA) were prepared by incubating with rat collagen I (50 μg/mL, Sigma Aldrich, Burlington, MA, USA) at 37 °C for 2 h then the excess collagen I solution was removed, and the inserts were dried for 30 min. Here, 2 × 10^5^ bEnd.3 cells were plated on the inserts and assembled into 6-well plates. The medium was added to the apical and basolateral compartments and incubated at 37 °C for 30 min before cell seeding. The cell confluence was achieved within 3 days of seeding. Then the assembly was subjected to oxygen-glucose deprivation treatment as previously described. The inserts were washed with Hanks’ balanced salt solution (HBSS) twice before being transferred to the other fresh 6-well plate. HBSS containing FITC-dextran (40 kDa, 1 mg/mL) was added into the upper chamber, and 1.5 mL of HBSS was added to the lower chamber in the incubator for 1 h at 37 °C Samples were collected from the basolateral compartment and were detected by Multimode microplate readers (Infinite 200, TECAN) with excitation and emission wavelengths of 485 nm and 530 nm, respectively.

### 4.8. TUNEL Assay

The bEnd.3 cells were plated on the coverslips in a 3.5 cm dish. Following the OGD-re treatment, as previously described, the coverslips were washed with phosphate buffer saline (PBS) then fixed in 4% paraformaldehyde for 30 min at room temperature (RT). The coverslips were permeabilized in 0.1% Triton X-100 in 0.1% sodium citrate for 2 min at 4 °C. Positive control for the TUNEL assay was incubated in 50 μL RQ1 RNase-free DNase (50 U/mL) for 10 min at RT. The positive control and treated coverslips were then washed in PBS and incubated with a 50-μL TUNEL reaction mixture, which contained a fluorescein reconjugated nucleotide mixture and the enzyme terminal deoxynucleotidyl transferase (TdT), prepared by following the manufacturer’s guidelines, for 1 h at 37 °C in the dark. The coverslip, which was not exposed to the TdT enzyme, would be a negative control. After washing with PBS, the coverslips were covered on microslides with a DAPI-Vectashield mounting medium in order to be examined with a DMI3000B inverted fluorescence microscope (Leica).

### 4.9. Measurement of Intracellular Reactive Oxygen Species (ROS) Levels Assay

According to the manufacturer’s protocol suggestion, in brief, 2.5 × 10^4^ cells were seeded on a dark and clear bottom 96-well microplate overnight at 37 °C. The medium was removed, and cells were washed gently with a 1× buffer. Then, the 1× buffer was aspirated, and a 100 μL/well diluted DCFDA solution was added to each well. The cells were incubated with a diluted DCFDA solution at 37 °C in the dark for 45 min. The DCFDA solution was removed, and a 100-μL 1× buffer was added to each well. The sample was read immediately with a multimode microplate reader (Infinite 200, TECAN), with excitation and emission wavelengths of 485 nm and 535 nm, respectively.

### 4.10. Immunochemistry and Double-Fluorescence Immunocytochemistry

After the mice brain sections (10-μm) cross sections had been blocked (1XPBS, 10% fetal bovine serum, 5% gelatin, and 0.1% Triton X-100) for 1 h, they were incubated with a mixture of two of the following primary antibodies: anti-CD31 (1:100, BD Pharmingen™, San Jose, CA, United States), anti-p75NTR (1:100, abcam, Burlingame, CA, USA ), anti-ZO1 (1:100, abcam, Burlingame, CA, USA), anti-VWF (1:100, abcam, Burlingame, CA, USA ) overnight at 4 °C. For immunochemistry, the brain slides were evaluated by immunochemistry through Dako kits (Dako EnVisionTM+ Dual Link System-HRP, Agilent, CA, USA). The procedure was conducted following the manufacturer’s instruction. The primary antibody was incubated with samples at 4 °C overnight followed by the HRP incubation for 1 h at room temperature. For immunofluorescence staining, the sections after primary antibody incubation were washed with 0.1 M PBS and then incubated with FITC- or Texas Red-conjugated anti-rabbit or anti-mouse IgG (1:400; Jackson ImmunoResearch, West Grove, PA, USA) for 1 h at room temperature then mounted with antifade mounting medium with DAPI (abcam, Burlingame, CA, USA). The IHC and fluorescence signals were detected and the results recorded using a microscope DMI300B (Leica, Wetzlar, Germany) and captured using Canon EOS550D with EOS Utility.

### 4.11. Statistical Analysis

Statistical analyses were performed using Graph-Pad Prism 5. The data were presented as the mean ± SEM. A paired t-test and one-way analysis of variance (ANOVA), followed by Tukey’s multiple comparison test, were used to evaluate the statistical significance. The western blot images were analyzed with the Quantity One 4.6.2 (Bio-Rad) for Windows. Statistical significance was determined when * *p* < 0.05.

## 5. Conclusions

Based on our presented results, the significance of our study reveal that brain vascular endothelial cells are highly sensitive to ischemic-hypoxic-induced injury and that AXT may be a potent remedy for stroke-induced brain injuries by ameliorating p75NTR elevation.

## Figures and Tables

**Figure 1 ijms-20-06168-f001:**
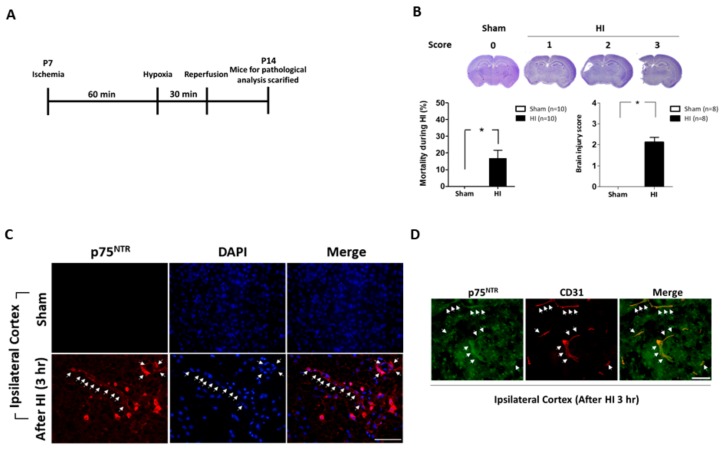
Establishing the ischemia-reperfusion mouse model. (**A**) Experimental scheme for ischemia-reperfusion. (**B**) Brain morphology observation stained and observed by Nissl stain. The mortality and brain morphology scores were also analyzed and quantified. (**C**) The representative immunofluorescent stain verifies p75 neurotrophin receptor (p75NTR) expression, the arrows indicating the p75NTR expression along the brain blood vessels. (**D**) The representative double-immunofluorescence staining in the hypoxic-ischemic (HI) group showed that the p75NTR-positive cells (arrowheads) and CD31-positive cells (arrowheads) were colocalized. (*n* = 8–10; values are mean ± SEM; * *p* < 0.05; scale bar: 100 μm).

**Figure 2 ijms-20-06168-f002:**
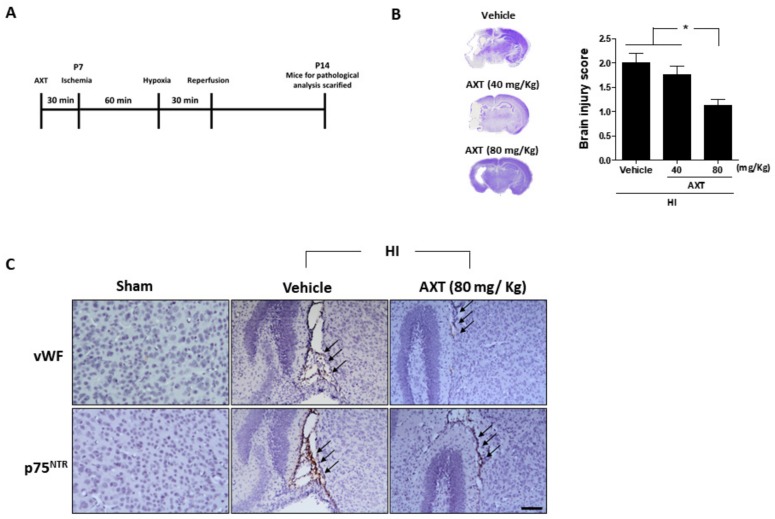
Analysis of astaxanthin (AXT) treatment for the ischemia-reperfusion mice and immunohistochemistry (IHC) brain slides. (**A**) AXT treatment experimental scheme for an ischemia-reperfusion mouse model. (**B**) Brain morphologies of mice treated with AXT, at 40 and 80 mg/kg, observed by Nissl staining and quantified. (**C**) The Von Willebrand factor (VWF), indicating endothelial cells and p75 neurotrophin receptor (p75NTR) expressions were observed by IHC staining in the mice brain slides. Arrows indicate the colocated sites of p75NTR and vWF (each group *n* = 14; values are mean ± SEM; * *p* < 0.05; scale bar: 100 μm).

**Figure 3 ijms-20-06168-f003:**
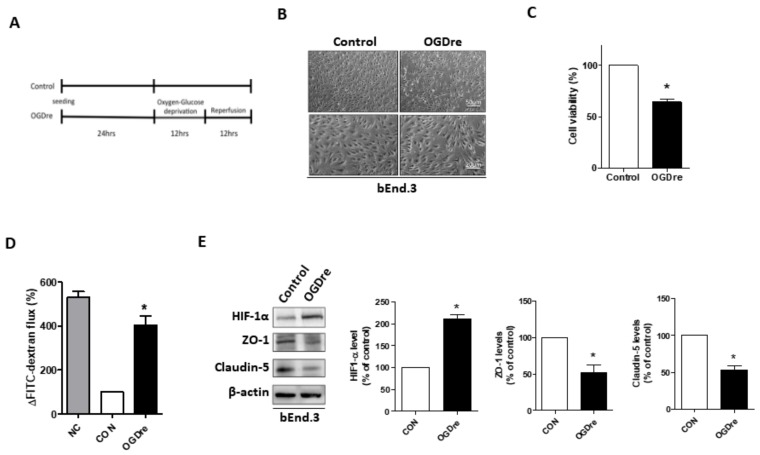
Establishment of the oxygen-glucose deprivation/reperfusion (OGDre) model using bEnd.3 cells and protein evaluation. (**A**) Experimental scheme for the ischemia-reperfusion cell model. (**B**) Morphologic alternations of monolayer formation in bEnd.3 cells after OGDre treatment was examined under a microscope. (**C**) The cell viability was examined by a 3-(4,5-dimethylthiazol-2-yl)-2,5-diphenyltetrazolium bromide (MTT) assay after OGDre treatment. (**D**) Endothelial monolayer permeability was examined by the detection of FITC-dextran after OGDre. NC: Negative control. (**E**) Protein expressions of HIF-1α, ZO-1, and claudin-5 were detected by western blot analysis. All the statistical results were compared to the control. (*n* = 3; values are mean ± SEM; * *p* < 0.05).

**Figure 4 ijms-20-06168-f004:**
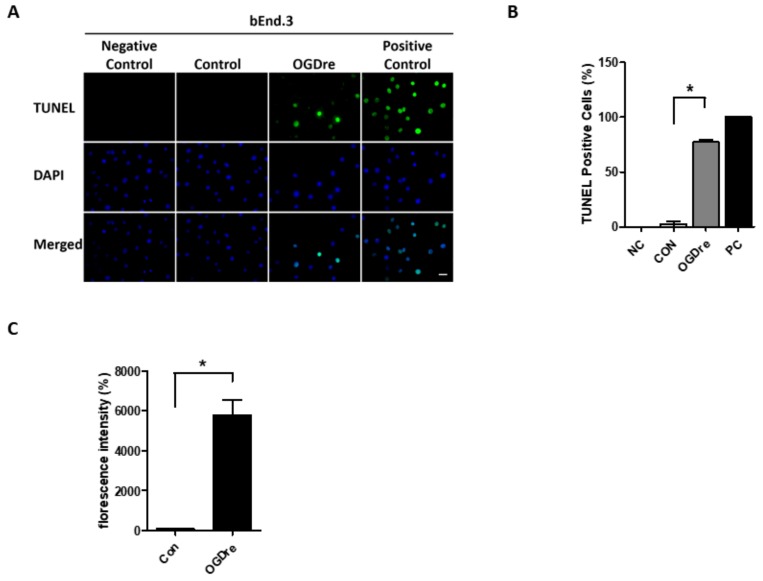
Apoptosis and reactive oxygen species (ROS) detection in bEnd.3 cells after oxygen-glucose deprivation/reperfusion (OGDre) treatment. (**A**) Apoptotic bEnd.3 cells were detected by a TUNEL assay. Apoptotic cells were marked with green fluorescence and the nuclei of cells were stained by blue fluorescence (DAPI). (**B**) Statistical analysis revealed a significant increase in TUNEL-positive cells in the OGDre groups. (**C**) ROS was detected by a DCFDA cellular ROS detection assay kit and the fluorescent values were quantified in both groups. (*n* = 5; values are mean ± SEM; * *p* < 0.05; scale bar: 10 μm).

**Figure 5 ijms-20-06168-f005:**
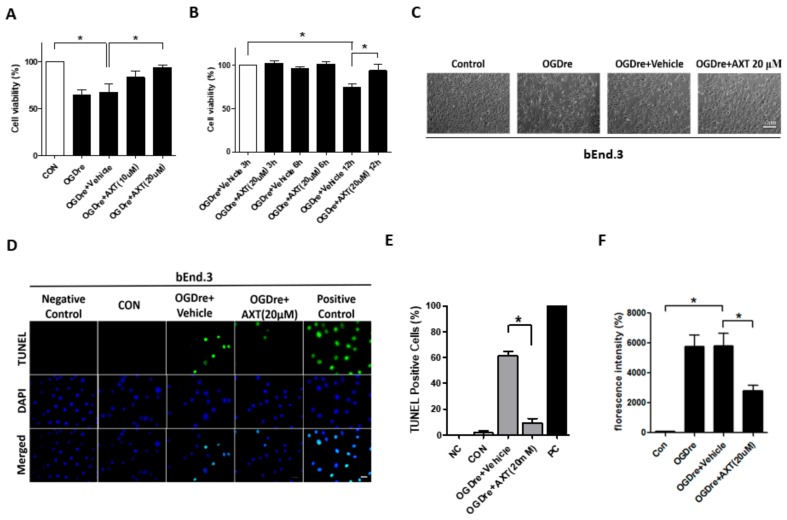
Astaxanthin (AXT) restored the viability of bEnd.3 cells after oxygen-glucose deprivation/reperfusion (OGDre) treatment. (**A**) Survival of bEnd.3 cells after OGDre was increased with AXT treatment in a dose-dependent manner. (**B**) Effects of AXT on bEnd.3 cell viability at different durations of reperfusion. (**C**) Morphologic alternations in bEnd.3 cells after OGDre with AXT treatment was examined under a microscope. (**D**) Apoptotic bEnd.3 cells were detected by a TUNEL assay. Apoptotic cells were marked with green fluorescence and the nuclei of cells were stained by blue fluorescence (DAPI). (**E**) Statistical analysis revealed a reduction of TUNEL positive cells after the treatment of AXT. (**F**) Reactive oxygen species (ROS) was detected with a DCFDA cellular ROS detection assay kit. (*n* = 5; values are mean ± SEM; * *p* < 0.05; scale bar 50 μm in (**C**) and 10 μm in (**D**)).

**Figure 6 ijms-20-06168-f006:**
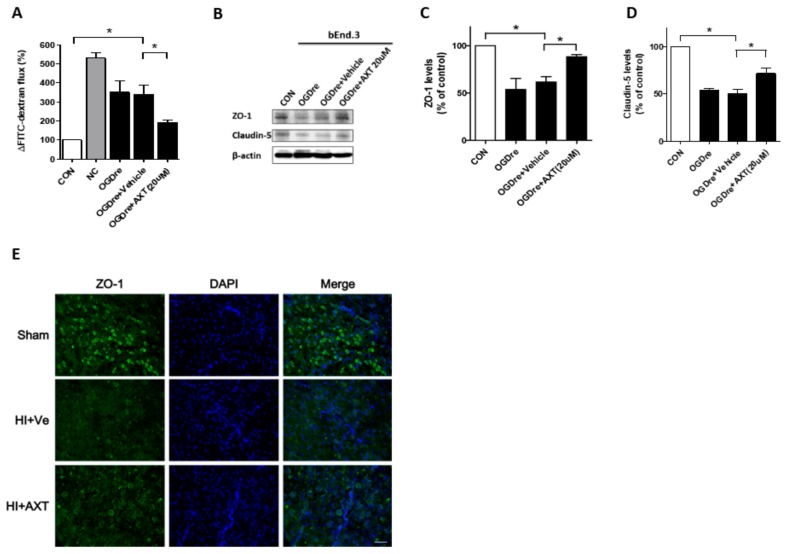
Expressions of tight junction protein after oxygen-glucose deprivation/reperfusion (OGDre) treatment with astaxanthin (AXT) in the bEnd.3 cell and HI animal models. (**A**) Endothelial monolayer permeability was examined by detection with FITC-dextran after OGDre with AXT treatment. (**B**) Expression of ZO-1 and claudin-5 were detected by western blot analysis. (**C**) Statistical quantification of ZO-1 expression level in OGDre compared with the control. (**D**) Statistical quantification of claudin-5 expression level in OGDre compared with the control. (**E**) The immunofluorescent staining of ZO-1 in HI mice brain. (*n* = 3; values are mean ± SEM; * *p* < 0.05; scale bar = 25 μm).

**Figure 7 ijms-20-06168-f007:**
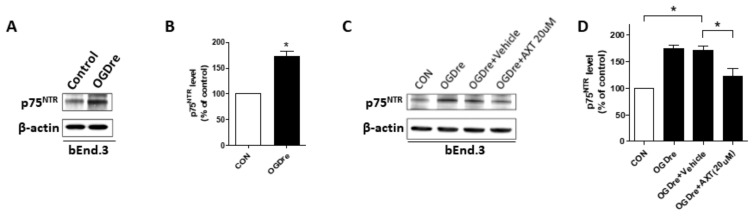
p75 neurotrophin receptor (p75NTR) expressions in bEnd.3 cell after oxygen-glucose deprivation/reperfusion (OGDre) with astaxanthin (AXT) treatment. (**A**) Expression of p75NTR was detected by western blot analysis after OGDre. (**B**) Statistical quantification of p75NTR expression level in OGDre compared with control. (**C**) Expression of p75NTR was detected by western blot analysis after OGDre with AXT. (**D**) Statistical quantification of p75NTR expression level in OGDre compared with control. (*n* = 3; values are mean ± SEM; * *p* < 0.05).

**Figure 8 ijms-20-06168-f008:**
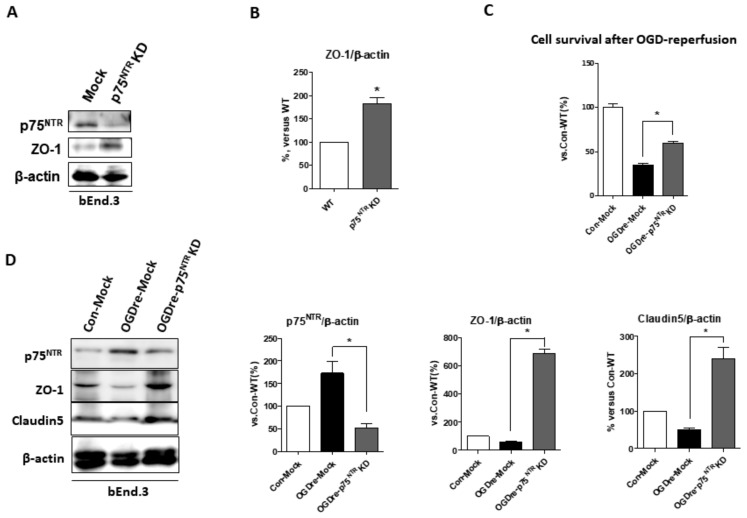
Cell viability and tight junction protein expressions of p75 neurotrophin receptor (p75NTR) knockdown of bEnd.3 cells after oxygen-glucose deprivation/reperfusion (OGDre) treatment. (**A**) Expressions of p75NTR and ZO-1 were detected by western blot analysis. (**B**) ZO-1 statistical quantification expression level compared with the control. (**C**) Cell survival in wild-type mock control and p75NTR knockdown cells after OGDre treatment. (**D**) Protein expressions of p75NTR, ZO-1, and claudin-5 were detected by western blot analysis. Statistical quantification expression level compared with control mock cells. (*n* = 5; values are mean ± SEM; * *p* < 0.05).

**Figure 9 ijms-20-06168-f009:**
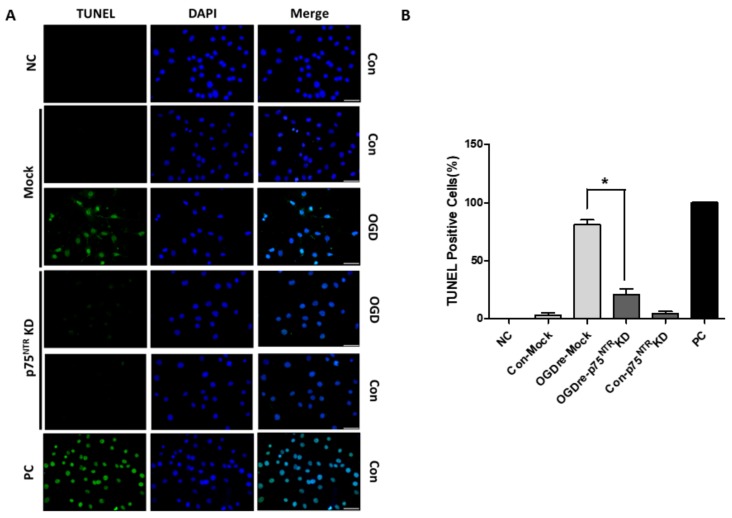
Apoptosis detection of mock and p75 neurotrophin receptor (p75NTR) knockdown after oxygen-glucose deprivation/reperfusion (OGDre) treatment. (**A**) Apoptotic cells were detected by a TUNEL assay. Apoptotic cells were marked with green fluorescence and the nuclei of cells were stained by blue fluorescence (DAPI). (**B**) Statistical quantification of TUNEL-positive signaling cells (*n* = 3; values are mean ± SEM; * *p* < 0.05; scale bar: 10 μm).

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
