# Peer review of "Astaxanthin Ameliorates Ischemic-Hypoxic-Induced Neurotrophin Receptor p75 Upregulation in the Endothelial Cells of Neonatal Mouse Brains"

_ijms, 2019, doi:10.3390/ijms20246168_

Round 1

Reviewer 1 Report

The manuscript covers experimental data on ischemic-hypoxic-induced p75NTR in vivo and vitro experiments. But there is too much data. Furthermore, there are 3 conclusions and the focus is unclear.

I think important experiments, the detect of ZO-1 and Claudin-5 protein in mice brain on Rice-Vannucci HI model, are missing.

Fig. 1 c, you should merge the signals of p75NTR and endothelial marker.

Reviewer 2 Report

The manuscript cover the data obtained in the experimental study of BBB breakdown caused by ischemic brain damage in vivo and OGD in vitro. I highly recommend to clarify the following issues in the current version of the manuscript:

The conclusion on p75NTR expression on BMECs is based just on single labeling of brain sections with the corresponding antibody. However, there are no data on double- or triple-labeling with some of the markers of BMECs to prove that p75NTR is really detected on cerebral endothelium but not in perivascular cells.  Fig. 1 shows that there was no p75NTR expression in the brain in sham-operated animals. However, this receptor should be constitutively expressed on neuronal cells. What is the reason for the absence of p75NTR staining in brain sections? The legend to the Fig. 2 indicates that there is co-expression of vWF and p75NTR, however, the quality/resolution is very poor, and no conclusion can be made.   Fig. 3B does not show any morphological alterations as stated in the legend. Fig. 3D and some other figures contains an abbreviation NC, but there is no explanation in the text.  Fig. 4C contains a typo.  The text requires English editing. The gropus of animals should be clearly described in the section "Materials and methods". List of references does not contain papers published in recent 5 years. What is a reason? 

Round 2

Reviewer 2 Report

The revised version contains appropriate corrections and additions.